# SARS-CoV-2 Serological and Biomolecular Analyses among Companion Animals in Campania Region (2020–2021)

**DOI:** 10.3390/microorganisms10020263

**Published:** 2022-01-24

**Authors:** Lorena Cardillo, Claudio de Martinis, Sergio Brandi, Martina Levante, Loredana Cozzolino, Luisa Spadari, Federica Boccia, Carmine Carbone, Marina Pompameo, Giovanna Fusco

**Affiliations:** 1Unit of Exotic and Vector-Borne Diseases, Istituto Zooprofilattico Sperimentale del Mezzogiorno, 80055 Portici, Italy; lorena.cardillo@izsmportici.it (L.C.); martina.levante@izsmportici.it (M.L.); loredana.cozzolino@cert.izsmportici.it (L.C.); 2Unit of Virology, Istituto Zooprofilattico Sperimentale del Mezzogiorno, 80055 Portici, Italy; sergio.brandi@cert.izsmportici.it (S.B.); giovanna.fusco@izsmportici.it (G.F.); 3Unit of Serology, Istituto Zooprofilattico Sperimentale del Mezzogiorno, 80055 Portici, Italy; luisa.spadari@izsmportici.it; 4Unit of Animal Health, Department of Prevention, Azienda Sanitaria Locale (ASL), Napoli 3 Sud, 80100 Naples, Italy; f.boccia@aslnapoli3sud.it (F.B.); c.carbone@aslnapoli3sud.it (C.C.); 5Unit of Animal Health “Presidio Ospedaliero Veterinario”, Department of Prevention, Azienda Sanitaria Locale (ASL), Napoli 1 Centro, 80100 Naples, Italy; marina.pompameo@aslnapoli1centro.it

**Keywords:** SARS-CoV-2, serology, real-time RT-PCR, dog, cat, surveillance

## Abstract

The first reports of SARS-CoV-2 among domestic and wild animals, together with the rapid emergence of new variants, have created serious concerns regarding a possible spillback from animal hosts, which could accelerate the evolution of new viral strains. The present study aimed to investigate the prevalence and the transmission of SARS-CoV-2 among both owned and stray pets. A total of 182 dogs and 313 cats were tested for SARS-CoV-2. Specimens collected among owned and stray pets were subjected to RT-PCR and serological examinations. No viral RNA was detected, while anti-N antibodies were observed in six animals (1.3%), one dog (0.8%) and five cats (1.7%). Animals’ background revealed that owned cats, living with owners with COVID-19, showed significantly different prevalence compared to stray ones (*p* = 0.0067), while no difference was found among dogs. Among the seropositive pets, three owned cats also showed moderate neutralizing antibody titers. Pets and other species are susceptible to SARS-CoV-2 infection because of the spike affinity towards their ACE2 cellular receptor. Nevertheless, the risk of retransmission remains unclear since pet-to-human transmission has never been described. Due to the virus’ high mutation rate, new reservoirs cannot be excluded; thus, it is reasonable to test pets, mostly if living in households affected by COVID-19.

## 1. Introduction

Since the first detection of SARS-CoV-2 in a dog in China and later among several animal species worldwide, concern has grown regarding their role in the transmission and maintenance of the virus, contributing to the pandemic scenario [1,2,3]. These reports sparked intense anxiety among society, especially due to the dissemination of fake news concerning pets as a source of human contagions, causing a dramatic intensification of abandonment cases [4]. Several studies have demonstrated the susceptibility of domestic and non-domestic animals to SARS-CoV-2, both in experimental and natural infections, mainly linked to close contact between infected owners/caretakers and susceptible species [3]. SARS-CoV-2 susceptibility is determined by the affinity between the viral receptor binding domain (RBD) of the spike glycoprotein and the host angiotensin converting enzyme 2 (ACE2). Vertebrates show ACE2-conserved primary structures that make them possible susceptible hosts [5]. The analysis of feline ACE2 has shown that SARS-CoV-2 can bind with high efficiency to the cat receptor [6]; indeed, it has been proven that this species can be infected both through natural and the experimental routes, shedding high viral titers, and an intra-specific contagion has also been demonstrated. On the other hand, lower susceptibility to SARS-CoV-2 in dogs, with low amounts of viral RNA shedding, has been observed [7,8,9,10]. Due to SARS-CoV-2’s error-prone nature during replication, several mutations have rapidly arisen that, in a few months, have led to the evolution of new variants, and some of them have rapidly spread worldwide [11]. Moreover, several studies have reported SARS-CoV-2 infections in companion animals by variants that appeared early in the pandemic [1,12,13], and later with B.1.1.7 Alpha [14,15] and the B.1.617.2 Delta variants [10].

Pets show mainly an inapparent course of infection; nevertheless, some authors have described symptomatic infected pets that showed mild digestive and respiratory symptoms [16]. A recent article has also reported infection with SARS-CoV-2 variant B.1.1.7 in dogs and cats with suspected myocarditis, with a history of close contact with COVID-19-positive owners [15]. Indeed, according to current knowledge, SARS-CoV-2 in animals can be mostly considered an anthropozoonosis, due to human-to-animal transmission [3,9]. Nevertheless, in March 2020, SARS-CoV-2 outbreaks in farmed minks in Denmark and the Netherlands [17,18], and later in Greece, Poland and the US, were reported [19,20,21,22]. It was hypothesized that minks were infected by positive farmers, followed by viral retransmission to persons working with infected minks [18,22]. These cases highlighted the importance of addressing the research on SARS-CoV-2 among susceptible animal species to evaluate ongoing or future possible spillover scenarios [23]. In particular, dogs and cats, which live in close contact with humans and inhabit the same environment, are highly exposed to human pathogens [24]. Thus, in order to assess the extent of SARS-CoV-2 infection among susceptible farmed and companion animals in a One Health approach, surveillance and monitory plans have been recommended [25]. For precaution principles, as well as for cognitive purposes, a “surveillance and prevention plan for SARS-CoV-2 in pets in Campania Region” has been issued, in response to the provisions of the Italian Ministry of Health reporting the “guidelines for the management of SARS-CoV-2 suspect pets” [26] and the suggestions of the European Commission and the World Organization for Animal Health (OIE) [27]. In this paper, we describe the results of active and passive surveillance among companion animals in the Campania region.

## 2. Materials and Methods

### 2.1. Sample Collection 

From April 2020 to October 2021, a total of 1341 samples from 495 pets (182 dogs and 313 cats) were analyzed for SARS-CoV-2 at the Istituto Zooprofilattico Sperimentale del Mezzogiorno (IZSM) of Portici, Naples (Southern Italy). The Local Health Authorities (AASSLL) and veterinary practitioners collected nasopharyngeal swabs, rectal swabs and serum samples from stray and owned pets, respectively, in compliance with active surveillance. Swabs were transferred to the IZSM in Universal Viral Transport Medium (UTM) (Copan, Brescia, Italy) and kept on ice until their arrival at the laboratories. For passive surveillance, 31 dead pets were provided by the AASSLL and subjected to necropsy by postgraduate veterinarians of the IZSM Veterinary Forensic Unit, in order to clarify the cause of death. During necropsy, in sterile conditions, heart, brain, liver, gut, spleen, lungs and tonsils were collected from each animal, together with pharyngeal, rectal and pulmonary swabs. Sample collection is summarized in Table 1.

### 2.2. Nucleic Acid Extraction

Tissue samples were subjected to homogenization and nucleic acid extraction in Biosafety Level 3 (BLS-3) laboratories. Approximately, 2 mg of organ tissue was collected in sterile conditions, placed in 2 mL tubes (Eppendorf, Hamburg, Germany) and suspended in 2 mL of sterile phosphate-buffered saline (PBS). Mechanical homogenization was performed with 0.5 mm glass beads (NextAdvance, Troy, NY, USA) using TissueLyser (Qiagen, Hilden, Germany) for 5 min at 30 Hz and subsequently clarified by centrifugation for 5 min at 1740× *g* (Eppendorf). Fecal swabs were processed as follows: approximatively 100 mg of feces was suspended in 900 µL of TissueLyser Buffer (ATL), vortexed on a mechanical shaker for 1–2 min at 40 Hz (VELP Scientifica, Usmate Velate, MB, Italy) and subsequently incubated for 10 min at room temperature to obtain the sedimentation of the macro-particles. Finally, 600 µL of supernatant was transferred into 1.5 mL tubes (Eppendorf) and incubated in a thermomixer (Eppendorf) at 70 °C for 10 min. Aliquots of 200 µL of supernatant were collected from each tissue homogenate, rectal swab and UTM, and MS2 phage was added as an internal process control (IC), included in the PCR kit (see below). Nucleic acid extraction and purification were performed using the QIAsymphony automated system with QIAsymphony DSP Virus/Pathogen Mini Kit (Qiagen), following the manufacturer’s instructions, eluted in 60 µL and finally stored at −80 °C until use.

### 2.3. SARS-CoV-2 Real-Time RT-PCR

The SARS-CoV-2 molecular investigation was carried out with the real-time multiplex RT-PCR technique in BLS-2 using the TaqPath COVID-19 CE-IVD RT-PCR Kit (Thermo Fisher Scientific), which allows the simultaneous amplification of 3 viral targets, the ORF1ab gene, encoding the polyprotein 1ab, FAM-labeled, the gene encoding nucleocapsid (N) protein, VIC-labeled and the gene encoding spike (S), ABY-labeled, and the internal process control, MS2 phage, JUN-labeled. The amplification was carried out in a final volume of 25 μL, including 5 μL of template, TaqPath 1-Step Multiplex Master Mix (4X) and a mix of specific primers and probes for the different investigated targets. The thermal profile included an initial Uracil-DNA glycosylase (UNG) incubation at 25 °C for 2 min, followed by a reverse transcription cycle at 53 °C for 10 min, an initial denaturation/enzymatic activation at 95 °C for 2 min, 40 denaturation cycles at 95 °C for 3 s and annealing/extension at 60 °C for 30 s. The amplifications were performed on a QuantStudio 5 real-time PCR system (Applied Biosystems, Foster City, CA, USA) thermal cycler.

### 2.4. Enzyme-Linked Immunosorbent Assay (ELISA) Antibody Test

Serum samples were tested to evaluate the presence of post-exposure IgG anti-SARS-CoV-2 nucleocapsid antibodies in dogs and cats using the ID Screen SARS-CoV-2 Double Antigen Multi-Species ELISA Kit (ID VET, Montpellier, France).

The indirect Enzyme-Linked Immunosorbent Assay (ELISA) was performed according to the manufacturer’s instructions. Briefly, 25 μL of each serum sample, positive and negative control was diluted in 25 μL of dilution buffer and incubated at 37 °C for 45 min. After washing three times, peroxidase-conjugated protein N recombinant antigen (HRP) was added and incubated at 25 °C for 30 min. After a second washing, substrate was added to each well and re-incubated at 25 °C for 20 min. Finally, stop solution was added to block the reaction and the optical density (OD) was read at 450 nm. The OD of each sample was calculated as a percentage Sample/Positive control (S/P%). According to the datasheet guidelines, samples with S/P% < 50% were considered negative, samples with S/P% between 50 and 60% were considered doubtful, while samples with S/P% > 60% were considered positive.

### 2.5. Quantitation of Serum-Neutralizing Antibodies

The micro-method serum virus neutralization test (SN) for the quantitation of anti-SARS-CoV-2 neutralizing antibodies was performed in four steps that included 2-fold serial dilution of the sera, addition of the reference virus and incubation, addition of cells and incubation and final reading with an inverted microscope.

Briefly, in the days preceding the SN, VeroE6 cells were cultured as described by the European Collection of Authenticated Cell Cultures [28], and viral titration of SARS-CoV-2 was performed according to the Spearman–Kärber method [29]. Working concentration 2000 TCID_50_/mL was realized by dilution in Eagle Minimum Essential Medium (EMEM) enriched with 2 mM L-glutamine and 1% antibiotic–antimycotic (Gibco, Life Technologies, Europe BV Bleiswijk, The Netherlands). Two different plates were prepared, one for tested sera and another for reference negative and positive sera, virus and cell controls.

Sera were heat-inactivated at 56 °C for 30 min and 10-fold pre-dilution was performed. Next, in a sterile 96-well flat-bottom microplate, 2-fold dilution was carried out, to obtain 1:10,240 final dilution. SARS-CoV-2 was added using 100 TCID_50_/50 μL (corresponding to 2000 TCID_50_/mL) and incubated at 37 °C ± 1 °C for 1 h in a microaerophilic atmosphere. At the end of the incubation, the cell suspension was added at a concentration to reach monolayer confluence in 24 h (approximately 1.5 × 10^5^ cells/mL) and incubated at 37 °C ± 1 °C for up to 72 h. The plate was first read at 24 h to evaluate the cytotoxic effect and the final reading was performed at 48 and 72 h for the cytopathic effect, by using the inverted microscope (Axioscope 5, Carl Zeiss, Oberkochen, Germany).

Detection of neutralizing antibodies was determined by the lack of cytopathic effect in VeroE6 cells at 72 h post-inoculation. The neutralizing antibodies titer was established on the basis of the highest dilution of serum that prevented infectivity. The results were expressed as positive when a titer ≥1/20 was observed and as negative with a titer <1/20.

### 2.6. Statistical Analysis

Prevalence was calculated at a 95% confidence level. All statistical analyses were performed by using SPSS software, version 24.0 (IBM Corporation). Results were considered statistically significant with a *p* value <0.05.

## 3. Results

Out of the 464 collected sera for active surveillance, six samples (1.3%) were found ELISA-positive for anti-SARS-CoV-2 N antibodies. Positive sera belonged to five cats (1.75%) and one dog (0.55%). Results are summarized in Table 2.

Animals’ background was investigated and no significant difference was found between owned and stray pets (*p* = 0.0934). On the other hand, higher prevalence was found in household cats (8.3%) with close contact with a recently confirmed COVID-19-positive owner (*p* = 0.0067). Among the four tested cats with a COVID-19-positive owner, three were found to be symptomatic. In particular, a 2-year-old intact female that showed gastro-enteric symptoms that spontaneously resolved within 15 days, a 15-year-old neutered male showed mild lethargy and loss of appetite and a 1-year-old intact female showed severe respiratory symptoms and died later because of respiratory distress. Unfortunately, necropsy and any other further investigations were not performed due to the owners’ lack of cooperation. The last two feline positive cases were a 14-year-old owned female and a 2-year-old stray female, captured by the Local Health Authorities, both sampled during minor surgeries. Finally, the only tested positive canine serum belonged to a 2.5-year-old mongrel stray male dog, captured in the same area as the stray cat. Gender evaluation showed no significant difference for both the cat (*p* = 0.5701) and dog (*p* = 0.7160) groups.

The ELISA-tested positive sera were then subjected to serum neutralization to assess the presence of neutralizing antibodies. Among six animals, positive results to SN were obtained in the three symptomatic owned cats, with 1:60, 1:80, 1:160 titers.

Finally, above all, the collected specimens for active surveillance from the 179 dogs and 285 cats, for a total of 796 samples, with 83 tissue samples for passive surveillance belonging to 3 dogs and 28 cats, no positive results were obtained in real-time RT-PCR.

## 4. Discussion

Reports of dog and cat infection have been described worldwide that have led to the development of monitoring and surveillance plans to better understand the infection rates as well as the morbidity of the pathogen in these species. Indeed, in compliance with the surveillance and prevention plan on SARS-CoV-2 among companion animals in the Campania region, the present study aimed to investigate, through active and passive surveillance, the prevalence and the transmission of the virus in stray and owned pets, as well as deceased animals. Therefore, from April 2020 to October 2021, 464 live and 31 deceased companion animals were analyzed for SARS-CoV-2. In agreement with data reported in other studies conducted among companion animals in the last two years, ranging between 0.2% and 14.7% [12,30,31], with differences related to the area under investigation and the period of the pandemic [32,33], we obtained an exposure rate of 1.3% but no evidence of viral RNA was observed when SARS-CoV-2 molecular analysis was performed, including animals from active surveillance coming from positive householders. Sampling time could justify these findings, since we were not able to rapidly test owned pets belonging to SARS-CoV-2-positive owners, due to the quarantine they were subjected to, according to current regulations, associated with the low viral shedding of the naturally infected asymptomatic pets [12,30]. Consequently, it is likely that the animals came into contact with the virus, seroconverted and eliminated the virus, which was also probably caused by the low viral excretion that these animal species exhibit [12]. In fact, the prevalence in these species is mainly based on serological investigation; indeed, the detection of viral RNA in pets is rarely reported, as observed in large-scale studies performed in Asia, Europe and North America [12,34]. In France, Temmam has not identified either antibodies or viral RNA [35], while Sailleau, in a study performed on 22 cats and 11 dogs, has identified a positive cat via RT-PCR [36]. Moreover, other Chinese studies have revealed a prevalence of 13.3% and 12% in dogs and cats, respectively [1,37]. Furthermore, Hamer and colleagues in Texas have detected SARS-CoV-2 in 15.3% of dogs and 47.1% of cats in pharyngeal and rectal swabs in a population of 76 companion animals from COVID-19-positive householders [38].

Although the canine ACE2 receptor (dACE2) computational analysis shows structural similarity with the human one (hACE2), higher susceptibility to SARS-CoV-2 is described in pet cats when compared to dogs [39]. It seems that the virus poorly replicates in dogs, probably due to the lower expression of ACE2 receptors in the respiratory tract [40], causing lower susceptibility in this species, where viral infection tends to be asymptomatic, with low viral excretion [41]. Nevertheless, our findings did not show a significant difference between cat and dog prevalence (*p* = 0.2930). On the other hand, besides the ELISA-tested positive sera, moderately high antibody titers were observed, demonstrating that cats develop a robust neutralizing antibody response [42].

Diseased pets are rarely reported, as SARS-CoV-2 experimental and natural exposure in these species can cause an asymptomatic/paucisymptomatic infection [9,43], but, when symptoms develop, sick pets show mild respiratory and gastro-enteric manifestations, as per our feline cases, such as fever, coughing, vomiting, diarrhea, lethargy, conjunctivitis and shortness of breath [2,38,44]; therefore, SARS-CoV-2 cases of pets, mostly cats, are similar to human COVID-19 [45].

We identified a significant difference between owned and stray cats (*p* = 0.0067), related to contact with confirmed COVID-19 human cases. In fact, it is reported that one of the main factors clearly influencing SARS-CoV-2 prevalence among companion animals [3,12,13,30] is living in infected households, with an eight-fold higher risk of testing seropositive [13]. Both in humans and in hamsters, gender has been identified as an important risk factor; indeed, male subjects with COVID-19 show more severe symptoms and higher fatality rates [46,47,48,49]. A large-scale study performed by Patterson and colleagues in Northern Italy revealed higher seroprevalence among male dogs, while no difference was observed among cats [12]. In contrast, no statistical evidence of a higher seropositive proportion of male pets was revealed by our findings, although, as they hypothesized, these findings could have been biased by the small number of positive animals [12]; thus, gender susceptibility in pets should be better clarified.

Collectively, all these results highlight the importance of conducting surveillance plans among companion animals. In fact, although the prevalence in pets could be considered insignificant compared to the scale of human infections [50], a relatively high seroprevalence in cats living in close contact with infected humans has been demonstrated, indicating the high anthropozoonotic potential of the virus. Moreover, it seems that pet ACE2 has increased affinity for variants carrying the 501Y mutation [41], which is currently spreading worldwide [51]; thus, it is fairly probable that the true prevalence in these species may be underestimated, also caused by the lack of investigation and the absence of symptoms that mainly characterize pet infection [30]. Finally, the horizontal transmission of the virus between cats and the evidence that SARS-CoV-2 is likely adapting to feline hosts could help the viral sequence evolution dynamics in this species [33].

## 5. Conclusions

Domestic animals are susceptible to SARS-CoV-2 infection and are able to develop neutralizing antibodies against this pathogen. Screening stray animals can help to better understand this poorly investigated aspect of the surveillance of SARS-CoV-2. On the basis of our results and other findings in the literature, there is currently no reason to justify a veterinary emergency regarding SARS-CoV-2 infection in pet animals, although this should not lead to a relaxation of surveillance among susceptible animal species, mostly in pets in close contact with COVID-19 cases. Further measures should be put in place to better clarify the role of these species in the transmission and maintenance of this virus, characterized by high morbidity and mutation rates.

## Figures and Tables

**Table 1 microorganisms-10-00263-t001:** Tested animals for active and passive surveillance.

		Dogs	Cats	Total
Active surveillance	Owned	101	48	149
	Stray	78	237	315
	Subtotal	179	285	464
Passive surveillance	Owned	2	7	9
	Stray	1	21	22
	Subtotal	3	28	31
Total		182	313	495

**Table 2 microorganisms-10-00263-t002:** Active surveillance SARS-CoV-2 risk assessment.

	Total	Positive (*n*)	Percentage (%)	*p* Value
Overall	464	6	1.29	
Species				
Cat	285	5	1.75	0.2930
Dog	179	1	0.55	
Background				
Owned	149	4	2.68	0.0934
Stray	315	2	0.63	
Cats				
Owned	48	4	8.33	0.0067 *
Stray	237	1	0.42	
Dogs				
Owned	101	0	0.0	0.4040
Stray	78	1	1.28	
Cats				
Female	194	4	2.06	0.5701
Male	91	1	1.09	
Dogs				
Female	67	0	0.0	0.7160
Male	112	1	0.89	

* Statistically significant (*p* < 0.05).

## Data Availability

The data presented in this study are included within the article.

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
