# Peer review of "SARS-CoV-2 Serological and Biomolecular Analyses among Companion Animals in Campania Region (2020–2021)"

_microorganisms, 2022, doi:10.3390/microorganisms10020263_

Round 1
Reviewer 1 Report
Recently, similar content has been reported. The authors must describe the differences between the reports previously reported so far and your findings.
In humans, males have a higher infection rate and aggravation rate of SARS-CoV-2 than females. It is well known that Sars-cov-2 is susceptible to cats and dogs. The authors should investigate whether there is a gender difference in SARS-CoV-2 infection in cats and dogs.
The authors should also consider SARS-CoV-2 variants in cats and dogs that are positive for SARS-CoV-2 infection. In the future, this study will provide us with important information to prevent the spread of COVID-19 infection.
Author Response
Issue 1. Recently, similar content has been reported. The authors must describe the differences between the reports previously reported so far and your findings.
Reply: Fixed as requested in conclusions. Finding of seroconversion in stray pets testifies to a viral circulation not due exclusively to contact with positive owners, but to a probable presence of the virus at environmental level (manure, food, indirect contacts).
Issue 2: In humans, males have a higher infection rate and aggravation rate of SARS-CoV-2 than females. It is well known that Sars-cov-2 is susceptible to cats and dogs. The authors should investigate whether there is a gender difference in SARS-CoV-2 infection in cats and dogs.
Reply: The issue has been fixed according to the reviewer request.
Issue 3: The authors should also consider SARS-CoV-2 variants in cats and dogs that are positive for SARS-CoV-2 infection. In the future, this study will provide us with important information to prevent the spread of COVID-19 infection.
Reply: In order to screen for variants using sequencing methods (Sanger, NGS), a low CT is recommended in addition to positivity. Unfortunately, attempts to obtain such outcomes from nasopharyngeal swabs have not been successful. Epidemiological surveillance on our territory is still in progress and our procedures include screening for variants.
Other requirements have been improved and fixed
Please see attachment: Revised Manuscript

Reviewer 2 Report
Cardillo et al. survey the presence of SARS-CoV-2 among domestic and wild animals and detect anti-N antibodies were observed in 6 animals. Domestic animals are therefore potentially susceptible to SARS-CoV-2 infection and are able to develop antibodies.
The overall content of the manuscript is fine; however, the scientific impact/importance seems rather low. Nonetheless, no further adaptations are required for publication.
Author Response
All requirements have been fixed, moreover other info have been included.
please find attached "Revised Manuscript"

Round 2
Reviewer 1 Report
The authors should add consideration to SARS-CoV-2 variants in the abstracts and introductions.
Author Response
Issue 1:
The authors should add consideration to SARS-CoV-2 variants in the abstracts and introductions
Reply:
The minor revision has been fixed as requested